# Estimation of Positions and Poses of Autonomous Underwater Vehicle Relative to Docking Station Based on Adaptive Extraction of Visual Guidance Features

Fengtian Lv [1,2,3,*], Huixi Xu [1,2,3], Kai Shi [1,2,3] and Xiaohui Wang [1,2,3]

1   State Key Laboratory of Robotics, Shenyang Institute of Automation, Chinese Academy of Sciences, Shenyang 110016, China; xhx@sia.cn (H.X.); shik@sia.cn (K.S.); wxh@sia.cn (X.W.)
2   Institutes for Robotics and Intelligent Manufacturing, Chinese Academy of Sciences, Shenyang 110169, China
3   Key Laboratory of Marine Robotics, Shenyang 110169, China
*   Correspondence: lvfengtian@sia.cn

**Abstract:** The underwater docking of autonomous underwater vehicles (AUVs) is conducive to energy supply and data exchange. A vision-based high-precision estimation of "the positions and poses of an AUV relative to a docking station" (PPARD) is a necessary condition for successful docking. Classical binarization methods have a low success rate in extracting guidance features from fuzzy underwater images, resulting in an insufficient stability of the PPARD estimation. Based on the fact that guidance lamps are blue strong point light sources, this study proposes an adaptive calculation method of binary threshold for the guidance image. To decrease the failure of guidance feature extraction, a guidance image enhancement method is proposed to strengthen the characteristic that the guidance lamps are strong point light sources with a certain area. The PPARD is estimated through solving the minimum value of the imaging error function for the vision-based extracted guidance features. The experimental results showed that the absolute estimation error for each degree of freedom in the PPARD was at most 10%, which was lower than that of the orthogonal iteration (OI) method. In addition, the proposed guidance feature extraction method proved to be better than the classical methods, with the extraction success rate reaching 87.99%.

**Keywords:** autonomous underwater vehicle; AUV docking; AUV pose; AUV position; guidance features

## 1. Introduction

The oceans occupy more than 70% of the earth's surface, and are rich in resources, including biological, mineral, and energy [1]. With the growing demand for resources by humans and sharp reduction in land resources, the exploration and development of ocean resources has become imminent. Autonomous underwater vehicles (AUVs) are widely used in ocean exploration. Compared to other types of underwater vehicles, AUVs have the advantages of a wide detection range, high mobility, and simple operation. AUVs are typically used in a variety of ocean exploration missions [2]. However, owing to their limited battery energy, the working time and capacity of AUVs are seriously limited. Future ocean exploration missions require AUVs to work for longer durations. A lack of energy may result in damages to AUV equipment and their motion, as well as errors or loss in perception data [3]. Therefore, as a technical solution to providing data transmission and exchange, energy augmentation, and maintenance of AUVs, underwater docking is very critical.

AUV guidance technology plays a decisive role in the successful docking of an AUV. It mainly comprises acoustic, visual, and electromagnetic guidance systems [4]. Acoustic guidance has a long effective distance and good versatility, but low precision. Electromagnetic guidance has high precision and a close effective distance, but poor versatility. Visual guidance has high precision, a close effective distance, and good versatility.

After an AUV completes its exploration operation, it usually uses the acoustic guidance technology with a wide array of effective ranges, from far homing to the initial docking position [5]. In the process of close docking, high-precision visual guidance technology is often adopted, usually by detecting guidance lamps of a docking station to estimate the "positions and poses of the AUV relative to the docking station" (PPARD) [6,7]. A high accuracy and robust estimation of PPARD is necessary to ensure a successful docking.

A vision-based estimation of PPARD mainly involves three steps: image preprocessing, guidance feature extraction, and PPARD calculation. Image preprocessing includes image enhancement and segmentation. Image enhancement is mainly based on image color correction and contrast enhancement to highlight the guidance features. The processing methods include the contrast limited adaptive histogram equalization (CLAHE) algorithm [8], median filter [9,10], and Gaussian filter [9]. Image segmentation is mainly used to segment guidance lamp regions to extract the guidance features. The processing methods, here, include the mean-shift algorithm [11], region growing method [12], and deep learning [13,14]. Guidance feature extraction can be divided into two types, namely guidance feature extraction based on image binarization (GFEBIB) and guided feature extraction based on edge detection (GFEBED). GFEBIB makes the guidance features white and other areas black by binarizing the image. Here, the main processing methods include the Otsu thresholding method [8], adaptive weighted Otsu method [12], self-tuning threshold method [15], and fixed thresholding [9,16]. GFEBED, on the other hand, requires guidance lamps to have a special shape, such as a heart shape. The guidance feature extraction is realized by extracting and judging the boundary shape of the object. The main processing methods include the Snake algorithm [11,12], Hough transform [17], etc. The PPARD is calculated according to the pixel coordinates and actual coordinates of the guidance features. Here, the main methods include direct linear transform (DLT) [15,18], efficient perspective-n-point algorithm (EPNP) [19], orthogonal iteration (OI) [8,20], and binocular vision [11,12].

However, the following problems need to be further researched. (1) With respect to underwater image enhancement, the existing image enhancement methods applied to guidance images cannot enhance the characteristics that the guidance lamps are strong point light sources. (2) Regarding the two-dimensional geometric feature extraction of guidance lamps, the previous studies have used classical binary methods to extract the guidance features. When the photographed environment or the camera pose and position change, the guidance lamps and their two-dimensional geometric feature extraction suffer from a low success rate. It is necessary to study the adaptive threshold binary method considering the guidance lamps as special color strong point light sources. (3) With regard to the PPARD estimation, measurement noise interference and guidance feature center coordinate deviation can cause a larger calculation error in the PPARD.

The dataset $D_{recovery}$ [14] was used to study the guidance feature extraction. To solve the above problems, a new method for PPARD estimation based on guidance image preprocessing and the adaptive extraction of visual guidance features was proposed in this study. The main contributions of this study are as follows: (1) an adaptive threshold binarization method for the guidance image was proposed based on the fact that the guidance lamps in the image are strong blue point light sources. The guidance features were extracted by combining image binarization and morphological processing; (2) a preprocessing method for the guidance image was proposed to enhance the intensity difference between the guidance lamps and the background area, and to segment the guidance target area. This improved the success rate of the guidance feature extraction from 68.89% to 87.99% for $D_{recovery}$; (3) based on the principle of camera imaging, this study also proposed an iterative optimization estimation model for PPARD estimation. The absolute estimation error for each degree of freedom (DOF) in the PPARD was less than 10%.

The remainder of this paper is organized as follows. Section 2 builds an estimation model for the PPARD based on the camera imaging principle. An adaptive threshold binarization method for the guidance image to obtain the input parameters for the estima-

tion model is described in Section 3. Section 4 presents the guidance image-preprocessing method. Section 5 discusses the experimental results. Finally, Section 6 concludes the paper.

## 2. Model Building for PPARD Estimation Based on Camera Imaging Principle

Figure 1 illustrates the coordinate system chosen for the docking station and camera. The docking station coordinate system (DCS) is expressed as *O-xyz*, the origin of which is located at the center of the docking station. The camera coordinate system (CCS) is expressed as $O_c$-$X_cY_cZ_c$, whose origin $O_c$ is located at the optical center of the lens, and the $Z_c$ axis coincides exactly with the optical axis. The origin of the image plane coordinate system, located at the top-left corner pixel of the image, can be expressed as $O_I$-*uv* (see Figure 1). The unit is pixel, and each point is represented by an integer.

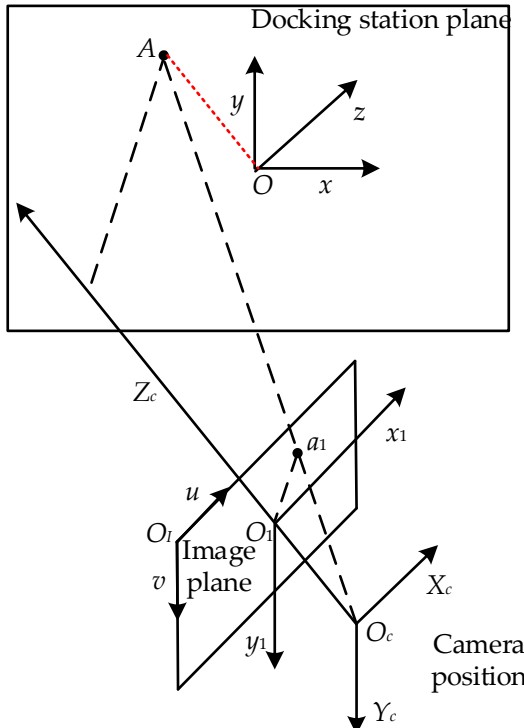

**Figure 1.** Coordinate systems for the docking station and camera.

The relationship between $O_I$-*uv* and *O-xyz* can be described as [21]:

$$\begin{bmatrix} u & v & 1 \end{bmatrix}^T = \frac{1}{Z_c}\boldsymbol{M}\begin{bmatrix} x & y & z & 1 \end{bmatrix}^T, \tag{1}$$

$$\boldsymbol{M} = \boldsymbol{F}\begin{bmatrix} \boldsymbol{R} & \boldsymbol{t} \\ \boldsymbol{0}^T & 1 \end{bmatrix} = \begin{bmatrix} m_{11} & m_{12} & m_{13} & m_{14} \\ m_{21} & m_{22} & m_{23} & m_{24} \\ m_{31} & m_{32} & m_{33} & m_{34} \end{bmatrix}, \tag{2}$$

where $\boldsymbol{F}$ is the camera interior parameter matrix; $\boldsymbol{R}$ is the rotation matrix; and $\boldsymbol{t}$ is the translation matrix of $O_c$-$X_cY_cZ_c$ relative to *O-xyz*.

Assume that the process of DCS transferring to CCS is as follows: (1) rotate around *x*-axis by $\theta_x$; (2) rotate around *y*-axis by $\theta_y$; (3) rotate around the *z*-axis by $\theta_z$; (4) move along the *x*-, *y*-, and *z*- axes by distances $l_x$, $l_y$, and $l_z$, respectively. The rotation matrix $\boldsymbol{R}$ and translation matrix $\boldsymbol{t}$ can be expressed as

$$\boldsymbol{R} = \left(\boldsymbol{R}_z\boldsymbol{R}_y\boldsymbol{R}_x\right)^T, \tag{3}$$

$$\boldsymbol{t} = \begin{bmatrix} l_x & l_y & l_z \end{bmatrix}^T, \tag{4}$$

where $R_x$, $R_y$, and $R_z$ are the basic rotation matrices for rotations around *x*-, *y*-, and *z*-axes, respectively.

Due to the fact that the camera is fixed on the AUV, if the " positions and poses of the camera relative to the docking station" (PPCRD) are estimated, the PPARD can be calculated.

Solving for the rotation transformation angle and translation distance of the CCS relative to the DCS, that is, rotation matrix $R$ and translation matrix $t$, is equivalent to the inverse process of camera imaging. Assume that the number of guidance feature points is $n$, and their coordinates in the DCS are $(x_i, y_i, z_i)$. Then, the following equation can be obtained:

$$\begin{bmatrix} \widehat{u}_i \\ \widehat{v}_i \\ 1 \end{bmatrix} = \frac{1}{Z_{ci}} F \begin{bmatrix} R_a & t_a \\ 0^T & 1 \end{bmatrix} \begin{bmatrix} x_i \\ y_i \\ z_i \\ 1 \end{bmatrix}, \tag{5}$$

where $(\widehat{u}_i, \widehat{v}_i)$ is the predicted pixel coordinate of point $(x_i, y_i, z_i)$ under the hypothetical rotation matrix $R_a$ and translation matrix $t_a$. The real pixel coordinates of the guidance feature points are $(u_i, v_i)$, which can be extracted through guidance image processing.

The imaging error function of guidance feature points can be built as follows:

$$e = \sum e_i = \sum \sqrt{\left(\widehat{u}_i - u_i\right)^2 + \left(\widehat{v}_i - v_i\right)^2}, \tag{6}$$

The following nonlinear programming problem is solved to estimate the PPCRD.

$$\begin{aligned} \min(e = \sum e_i) \\ s.t. \, t_l < x < t_u \end{aligned} \tag{7}$$

$$x = \begin{bmatrix} \theta_x & \theta_y & \theta_z & l_x & l_y & l_z \end{bmatrix}, \tag{8}$$

where $t_u$ and $t_l$ are the upper and lower limits of the PPCRD, respectively. During the docking process, the AUV will first use acoustic information to adjust the docking positions and poses. A visual guidance strategy is adopted when the distance between the AUV and the docking station is less than 10 m. There will not be too much deviation of PPCRD. Therefore, $t_u$ and $t_l$ were selected as $[-30°, -30°, -30°, -1500 \text{ mm}, -1500 \text{ mm}, -10{,}000 \text{ mm}]$ and $[30°, 30°, 30°, 1500 \text{ mm}, 1500 \text{ mm}, 0]$, respectively, in this paper. The interior-point method [22] was used to solve Equation (7), which is the estimation model for the PPCRD. To solve for the PPCRD, the real pixel coordinates of the guidance feature points need to be extracted.

## 3. Guidance Feature Extraction Based on Adaptive Threshold Image Binarization

The guidance image for dataset $D_{recovery}$ is shown in Figure 2a. The resolution of guidance images in the dataset is $640 \times 480$. Owing to the different absorption capacities of water for different wavelengths of light, the entire image was rendered green. The green component G of the image is shown in Figure 2b. At this time, it was difficult to extract the guidance lamps from component G by using the threshold method. The guidance lamps were blue in color. Therefore, the blue component B was used to extract the guidance features. Component B of the guidance image is shown in Figure 2c. It can be seen that the *B* values of the guidance lamps were significantly higher than those of the background area.

The pixel coordinates in the image are represented by $(u, v)$, and the corresponding *B* value is represented by $b(u, v)$. With a one-pixel resolution, component B was mapped onto a three-dimensional surface, as shown in Figure 3. There were eight peaks and troughs in the guidance lamp area in the figure. The maximum *B* value of the guidance lamps and corresponding position of the value could be obtained by extracting the position and value of the peaks. Each peak position could be regarded as the position of the guidance

lamps. The extraction process of the guidance lamp peaks in the guidance image is shown in Figure 4, and the extraction results are shown in Figure 5.

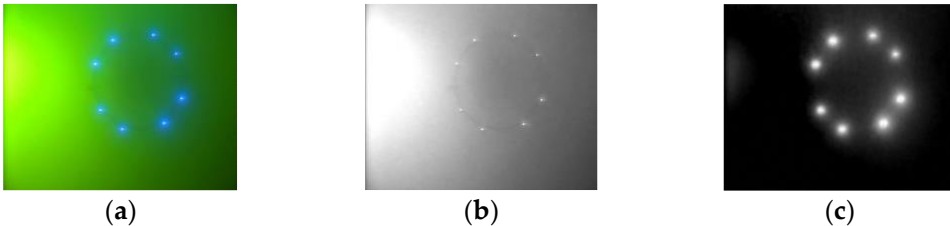

| (**a**) | (**b**) | (**c**) |

**Figure 2.** Guidance image: (**a**) original image; (**b**) green component G of the guidance image; (**c**) blue component B of the guidance image.

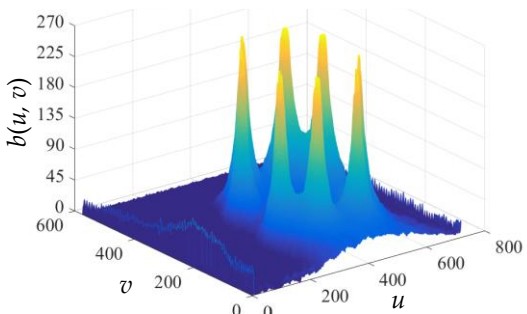

**Figure 3.** Three-dimensional surface of component B.

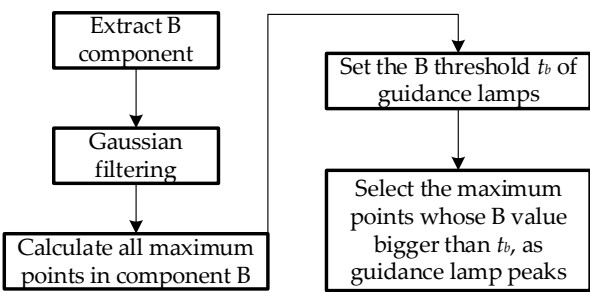

**Figure 4.** Extraction process of guidance lamp peaks.

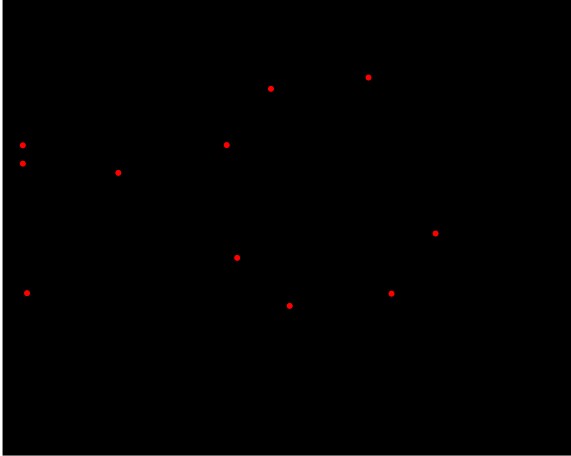

**Figure 5.** Extraction result of guidance lamp peaks.

The following problems can be observed from the results.

(1)  Noise points extracted as guidance lamp peaks for filtering could not guarantee the smoothness of the surface in Figure 3. By extracting the wave peaks, a large number of connected areas of the wave peaks could be obtained, which was much larger than the number of guidance lamps.

(2)  In order to test whether the noise can be eliminated by setting the wave peaks extraction threshold $t_b$, it is gradually increased from a small value (100) to 200. During this period, the noise cannot be eliminated. However, when $t_b$ is 200, a guidance lamp is eliminated. As shown in Figure 5, not all guidance lamps could be extracted, and there were noise points in the figure. Therefore, all the guidance lamps could not be extracted without noise by adjusting $t_b$.

(3)  The area of the extracted connected area was very small and close to a point. Therefore, morphological processing could not be used to remove the noise. This was not conducive to the location of the guide lamps.

To address the above problems, binarization of the image was considered to obtain a larger connected area of the guidance lamps. The binarization threshold of the image was selected according to the value of the wave peaks and wave troughs, as shown in Figure 3. The threshold $t$ is

$$t = (p_{\max} - v_{\max})/2 + v_{\max}, \tag{9}$$

where $p_{\max}$ is the maximum value of all peak values, and $v_{\max}$ is the maximum of all the wave valley values.

Component B of the guidance image was binarized according to threshold $t$. The pixels with a *B* value larger than $t$ were assigned a value of 1, whereas others were assigned 0, i.e.,

$$f_b(u,v) = \begin{cases} 1 & b(u,v) \geq t \\ 0 & b(u,v) < t \end{cases}, \tag{10}$$

where $f_b(u, v)$ is a binary image. The guidance image was binarized by Equation (10), and then morphological processing was performed to extract the guidance features, as shown in Figure 6.

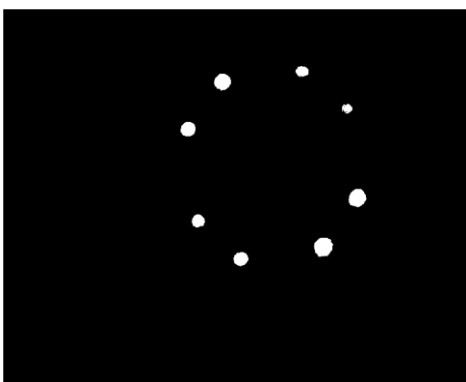

**Figure 6.** Extraction result of guidance features.

## 4. Preprocessing Method of Underwater Guidance Images

### 4.1. Underwater Guidance Image Enhancement

As the guidance lamps were blue in color, and the light spread gradually, when the AUV stood far away from the docking station, variations in the photo camera poses/positions caused the areas among adjacent lamps to be blue, as shown in Figure 7a,b. This would lead to a binary threshold value that was too high or too low and resulted in a loss of guidance lamps or some adjacent guiding lamps merging together or overlapping each other, as shown in Figure 7c,d.

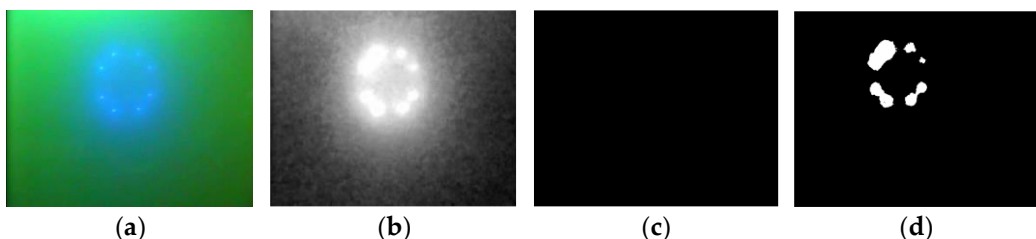

**Figure 7.** Problems of guidance lamp extraction caused by variations in camera poses/positions: (**a**) original image; (**b**) blue components; (**c**) loss of guidance lamps; (**d**) adjacent guiding lamps merging together or overlapping each other.

To solve the above problems, the following requirements must be met through image enhancement:

(1)    Adjacent guidance lamps in the image should be clearly separated, that is, the values of peaks and troughs of any two adjacent lamps should differ significantly.
(2)    Guidance lamps should present the characteristics of strong light sources.
(3)    The highlighted area of guiding lamps should have an adequate area, not just a point, to prevent morphological filtering and causing the loss of guiding lamps.
(4)    The brightness of all the guidance lamps should be similar.

Figure 8 shows the red and green components of the guidance image. Compared with the background area, the guidance lamps in both the images present a bright spot. Thus, the guidance lamps met requirements (1) and (2).

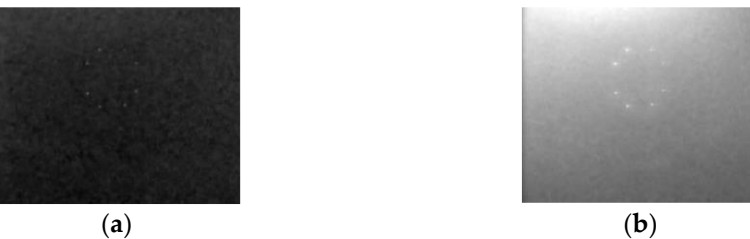

**Figure 8.** Red and green components of the guidance image: (**a**) red component; (**b**) green com-ponent.

Owing to a severe absorption of red light by water, the brightness of the red component was low. At the same time, owing to the influence of external light, the intensity of the green component was high in the no-guidance lamp area. To solve these problems, the Retinex algorithm was used to process the image, which is given by [23].

$$I = I_R \cdot I_L, \tag{11}$$

where $I$ is the image obtained by the camera; $I_R$ is the incident component; and $I_L$ is the reflection component. $I_R$ is mainly determined by the incident light, and $I_L$ is determined by the reflection property of the object itself. The Retinex algorithm was used to obtain the reflection images $R_L$ and $G_L$ of the red and green components, respectively, as shown in Figure 9.

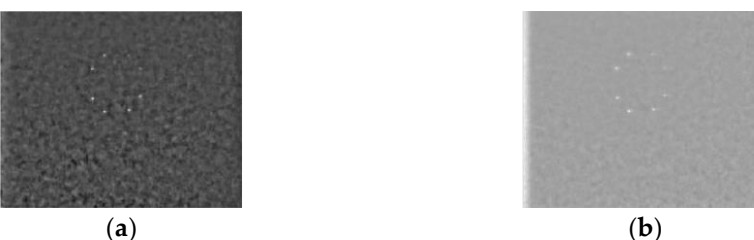

**Figure 9.** Reflection images of red and green components: (**a**) reflection image of red components; (**b**) reflection image of green components.

In order to meet the above four requirements after the image processing, the reflected images of R and G components were extracted using the Retinex algorithm to eliminate the interference of ambient light. The guidance lamp was a blue point-light source. To maintain the characteristics of the guidance light, the Retinex algorithm was not used for the B component to maintain its blue component characteristics.

The images $R_L$ and $G_L$ were added to further strengthen the guidance lamps, which had the characteristic of being strong light sources. Then, image $K$ was obtained, as shown in Figure 10.

$$K = R_L/2 + G_L/2, \tag{12}$$



**Figure 10.** Image $K$ ($K = R_L/2 + G_L/2$).

A dynamic piecewise nonlinear adjustment was used to enhance $K$ and improve the enhancement effect. The background region intensity was adjusted by an exponential function of index of 2 to lower its value and fasten to a low value closer to 0. The guidance lamp region intensity was adjusted using an exponential function of index of 0.5, to smoothen and narrow its value. The adjustment equation is as follows:

$$K' = \begin{cases} K'_{wt} \times \left(\frac{K}{K_{wt}}\right)^2 & 0 \le K < K_{wt} \\ K'_{wt} + (K'_{tb} - K'_{wt}) \cdot \sin\left[\left(\frac{K - K_{wt}}{K_{tb} - K_{wt}}\right) \times \frac{2\pi}{3}\right] & K_{wt} \le K < K_{tb}, \\ K'_{tb} + (1 - K'_{tb}) \times \sqrt{\left(\frac{K - K_{tb}}{1 - K_{tb}}\right)} & K_{tb} \le K \le 1 \end{cases} \tag{13}$$

$$\exists! \varepsilon_i \in \varepsilon, \sum_{S=0}^{\varepsilon_{i-1}} N_S < 0.5N \wedge \sum_{S=0}^{\varepsilon_i} N_S \ge 0.5N \Rightarrow K_{wt} = \varepsilon_i, \tag{14}$$

$$\exists! \varepsilon_j \in \varepsilon, \sum_{S=0}^{\varepsilon_{j-1}} N_S < 0.95N \wedge \sum_{S=0}^{\varepsilon_j} N_S \ge 0.95N \Rightarrow K_{tb} = \varepsilon_j, \tag{15}$$

$$\varepsilon = \{a/255 | a \in \mathbb{Z} \& 0 \le a \le 255\}, \tag{16}$$

where $N$ is the total number of pixels of the entire image and $N_S$ is the pixel number of value $S$. The parameters $K_{wt}$ and $K_{tb}$ are the adjusted upper limits of the intensity value of image $K$. Figure 11 shows the image intensity dynamic piecewise nonlinear adjustment results.

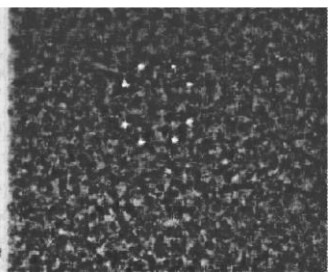

**Figure 11.** Results of image intensity dynamic piecewise nonlinear adjustment (image $K'$).

It can be seen that the brightness of the guidance lamps was enhanced and the area of the highlight-connected area of the guidance lamps increased. However, there were some high-brightness noises in the image. The binary processing could easily produce the pseudo-guidance lamps. To solve this problem, the component B was adjusted using Equation (17) to obtain $B'$. Then, $B'$ and $K'$ were added to obtain a new image $M$, as shown in Figure 12.

$$B' = \begin{cases} B'_{wt} \times \sqrt{\frac{B}{B_{wt}}} & 0 \leq B < B_{wt} \\ B'_{wt} + (B'_{tb} - B'_{wt}) \cdot \cos\left[\left(\frac{B-B_{wt}}{B_{tb}-B_{wt}}\right) \times \frac{2\pi}{3}\right] & B_{wt} \leq K < B_{tb}, \\ B'_{tb} + (1 - B'_{tb}) \times \sqrt{\left(\frac{B-B_{tb}}{1-B_{tb}}\right)} & B_{tb} \leq B \leq 1 \end{cases} \tag{17}$$

$$\exists! \varepsilon_i \in \varepsilon, \sum_{S=0}^{\varepsilon_{i-1}} N_S < 0.5N \wedge \sum_{S=0}^{\varepsilon_i} N_S \geq 0.5N \Rightarrow B_{wt} = \varepsilon_i, \tag{18}$$

$$\exists! \varepsilon_j \in \varepsilon, \sum_{S=0}^{\varepsilon_{j-1}} N_S < 0.8N \wedge \sum_{S=0}^{\varepsilon_j} N_S \geq 0.8N \Rightarrow B_{tb} = \varepsilon_j, \tag{19}$$

$$M = K'/2 + B'/2, \tag{20}$$

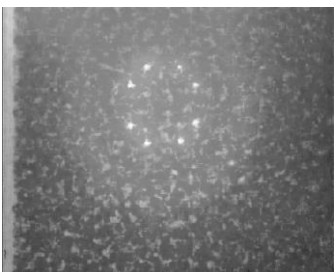

**Figure 12.** The final enhancement image $M$.

The brightness of the guidance lamps and noise had an evident difference. Image $M$ met the four requirements mentioned above and was more advantageous for image binarization processing and the feature extraction of guidance lamps.

*4.2. Segmentation of Guidance Lamp Region*

The image was divided into $n$ image blocks and the following features were extracted for each image block:

(1)     Average intensity $l_m$;
(2)     Gradient mean $d_m$;
(3)     Strong boundary pixels number $b$.

The mean values of all of the image blocks of the three features were expressed as, $l_{mm} = \sum l_m/n, d_{mm} = \sum d_m/n$, and $b_m = \sum b/n$. The image was segmented according to Equation (21) to obtain the guidance lamp area, as shown in Figure 13.

$$M_{new}(x,y) = M(x,y) \times S(x,y), \tag{21}$$

$$S(x,y) = \begin{cases} 1 & l_{mi} \geq l_{mm} \& d_{mi} \geq d_{mm} \& b_i \geq b_m \& (x,y) \in \Omega_i \\ 0 & else \end{cases}, \tag{22}$$

where $l_{mi}$, $d_{mi}$, and $b_i$ are the above three features of the image block $\Omega_i$. After image preprocessing, the guidance features were extracted from the image $M_{new}$ (instead of the component B) using the adaptive threshold image binarization method proposed in Section 3. The results are shown in Figure 14.

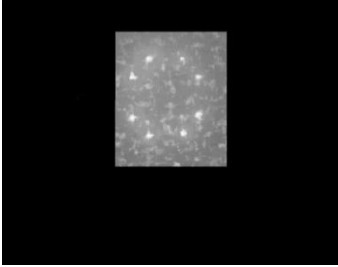

**Figure 13.** Result from segmentation.

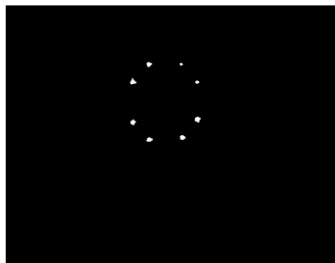

**Figure 14.** Guidance features extracted from preprocessed image $M_{new}$.

## 5. Experimental Verification

### 5.1. PPARD Estimation Steps Based on Image Processing

The following are the PPARD estimation steps:

(1)   Acquire a guidance image from the camera.
(2)   Preprocess the guidance image, including image enhancement and image segmentation.
(3)   Binarize the image using the adaptive threshold binarization method, proposed in Section 3. Extract the guidance features and calculate their pixel coordinates.
(4)   Input the pixel coordinates and actual coordinates of the guidance features into Equation (7) to solve for PPARD.

### 5.2. Experimental Equipment

Due to the difficulty in obtaining the real relative PPARD, it is impossible to use the camera mounted on AUV to conduct a PPARD estimation accuracy test. Thus, a PPARD estimation test platform (shown in Figure 15) was designed to analysis the accuracy of PPARD estimation in this paper. The experimental device was rectangular-shaped. Four guidance lamps were installed at the midpoint of the four sides at one end of the experimental device to simulate the entrance of the docking station. The other end was equipped with a camera to simulate the camera installed in front of the AUV platform. The two ends were connected by rods with adjustable lengths to realize the distance adjustment in the *z*-direction.

The camera, whose resolution is 640 × 480, was installed on a three-dimensional turntable. It could rotate around the *x*-, *y*-, and *z*- axes. A three-dimensional turntable was installed on the mounting plate of the camera. Several camera mounting positions with different *x*- and *y*- coordinate combinations were designed on the camera mounting plate, which could help to adjust the camera's positions in *x*- and *y*- directions. The experimental device could adjust and sign the six DOFs of the camera relative to the DCS *O-xyz*. The PPARDs were fixed on the land. Then, the experimental device was placed in the pool (see Figure 16) to collect the underwater guidance image. The positions and poses of the camera relative to the DSC *O-xyz* were estimated to test and analyze the accuracy of the method proposed in this paper.

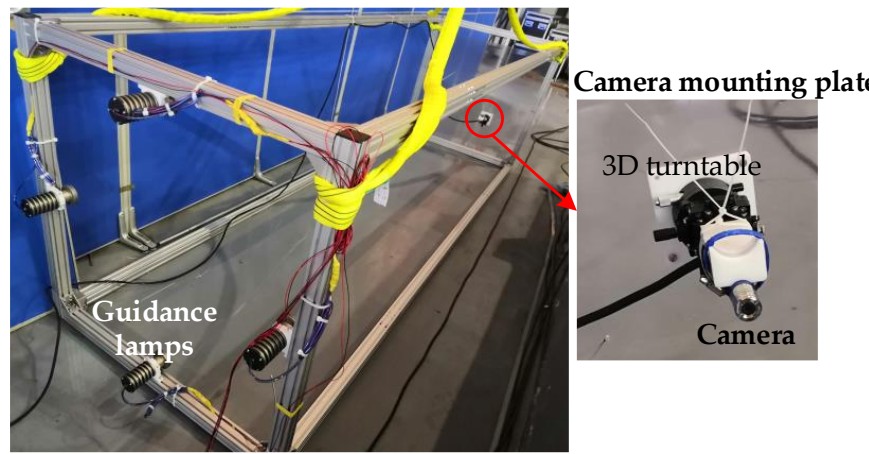

**Figure 15.** PPARD estimation test platform used in the study.

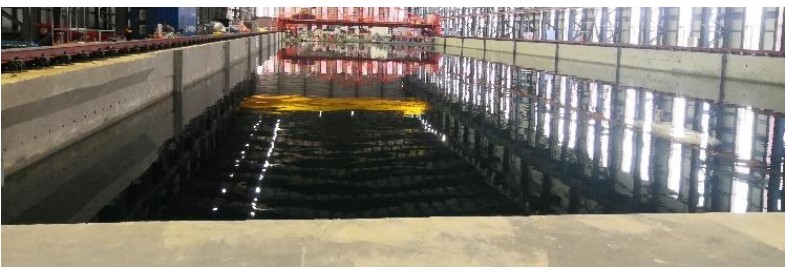

**Figure 16.** Photograph of the pool used in the study.

The values of six DOFs set in the experiment are listed in Table 1. If all of the combinations for poses and positions were tested, the number of experiments would be very large. Thus, the orthogonal experiment method was used to remove the redundant experimental PPARD combinations, and a total of 25 experimental groups were used.

**Table 1.** Values of six DOF set in the experiment.

| DOF | Value |
| :---: | :---: |
| $\theta_x$ (°) | −10, −5, 0, 5, 10 |
| $\theta_y$ (°) | −10, −5, 0, 5, 10 |
| $\theta_z$ (°) | −10, −5, 0, 5, 10 |
| $l_x$ (mm) | −400, −200, 0, 200, 400 |
| $l_y$ (mm) | −400, −200, 0, 200, 400 |
| $l_z$ (mm) | −2000, −3000, −4000 |

*5.3. Accuracy Analysis of PPARD Estimation*

Figure 17a shows an example of a guidance image collected by the camera. In the experiment, the PPARD for each group of experiments was marked according to the designed position and the angle adjusted by the three-dimensional turntable. Figure 17a is the original image, whereas Figure 17b shows the extracted guidance features, and Figure 17c shows the result of the PPARD estimation.

Figure 18 shows the PPARD estimation results. It can be seen that the estimated values of the six DOFs fluctuated around the experimentally set values. To obtain the absolute accuracy of the PPARD estimation, the differences between the estimated and set values were calculated, and are shown in Figure 19.

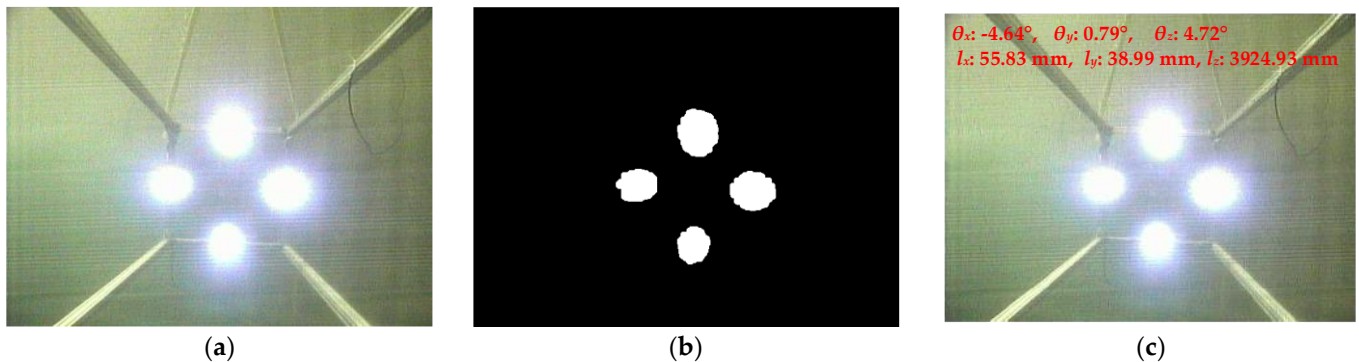

**Figure 17.** Example of PPARD estimation: (**a**) original image; (**b**) extracted guidance features; (**c**) PPARD estimation result.

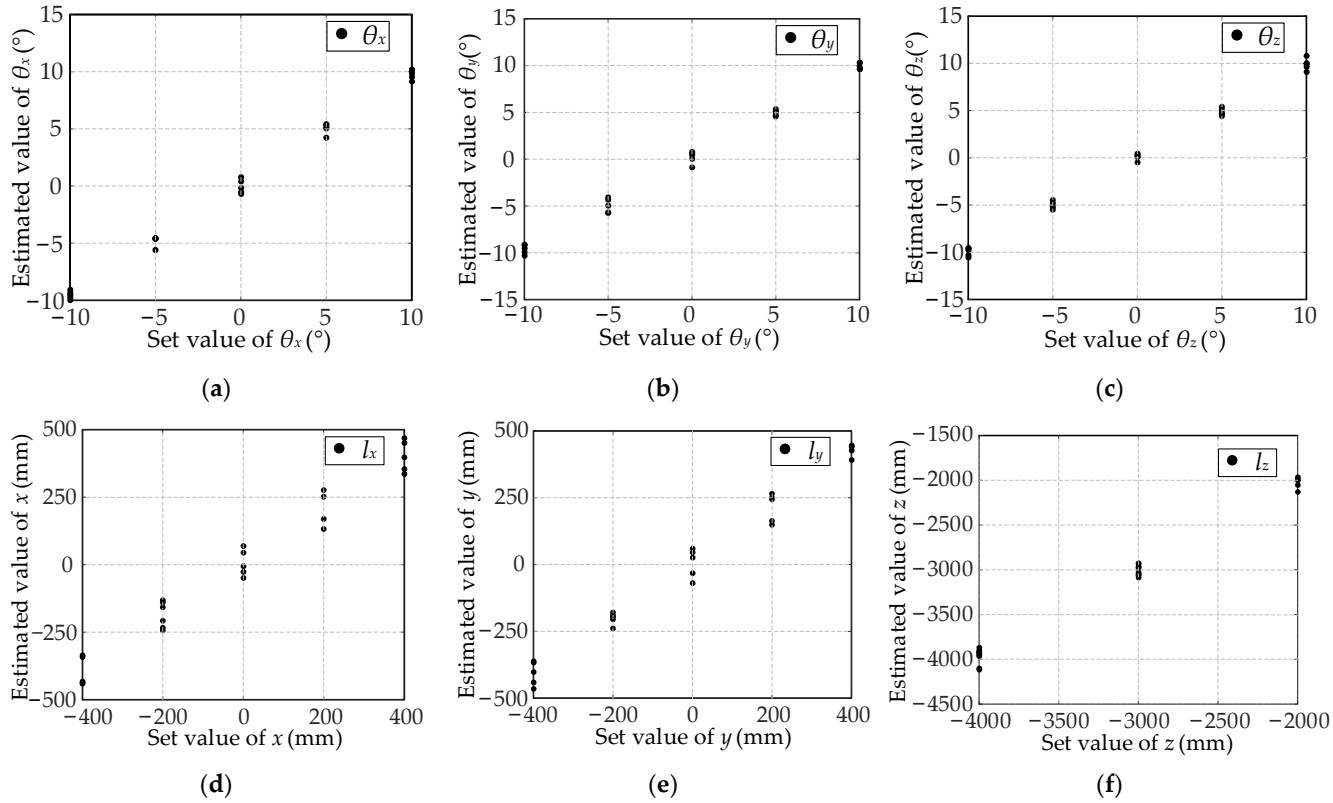

**Figure 18.** PPARD estimation results: (**a**) $\theta_x$; (**b**) $\theta_y$; (**c**) $\theta_z$; (**d**) $l_x$; (**e**) $l_y$; (**f**) $l_z$.

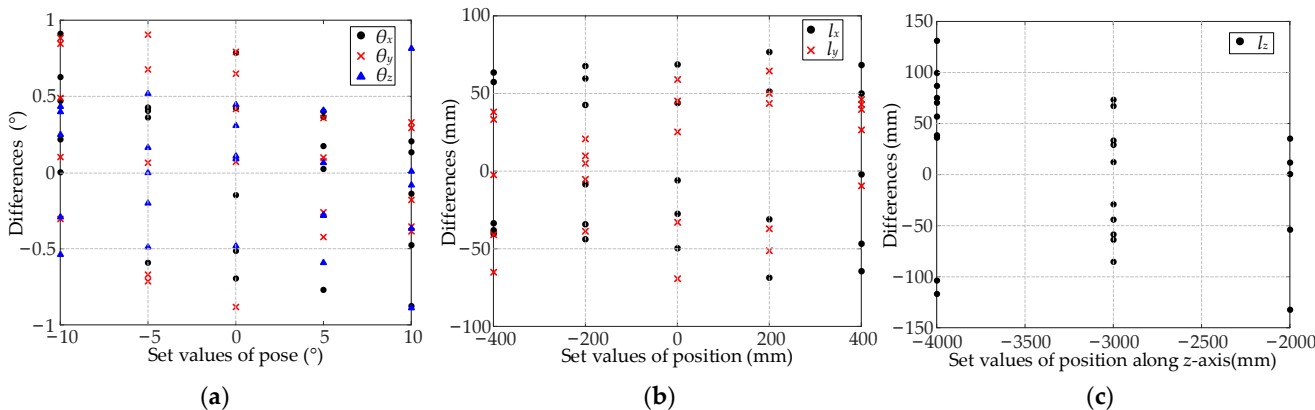

**Figure 19.** Differences in the estimated and set values: (**a**) three rotation angles; (**b**) $l_x$ and $l_y$; (**c**) $l_z$.

It shows that, for any DOF, the errors under different positions or poses setting values were relatively close. The estimation error for the rotation angles of the three axes was less than 1° and the relative error was less than 10%. The estimation error of the positions in the *x*- and *y*- directions was less than 80 mm, and the relative error was less than 10%. The estimation error of the positions in the *z*-direction was less than 140 mm and the relative error was less than 3.5%.

To further evaluate the accuracy of the PPARD estimation, the evaluation methods proposed for rotation error $E_{rot}$ and translation error $E_{trans}$ in [24] were used.

$$E_{rot} = \max_{k=1}^{3} \arccos\left(r_{true}^{k} \cdot r^{k}\right), \tag{23}$$

$$E_{trans} = \|t - t_{true}\| / \|t_{true}\| \times 100\%, \tag{24}$$

where $r_{true}^{k}$ and $r_{k}$ are the *k*-th columns of the true value $R_{true}$ and the estimated value $R$ of the rotation matrix; $t$ and $t_{ture}$ are the estimated and true values of the translation matrix, respectively. Using these two indicators to evaluate the accuracy of the PPARD estimation algorithm, the results are shown in Figure 20. In the experiment, the distance between the camera and guidance lamps along the *z*-direction was 4000 mm for experiments 1–10, 3000 mm for experiments 11–20, and 2000 mm for experiments 21–25. From the results, it can be seen that the rotation error was less than 1.3°. The general trend was that the rotation error decreased with a decrease in the distance between the camera and guidance lamps along the *z*-axis. The translation error was less than 12%.

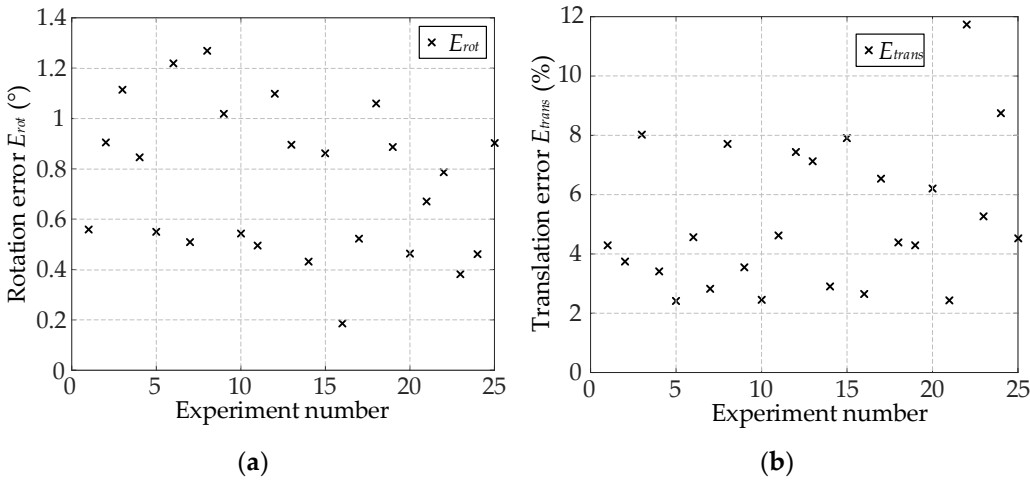

(a)                     (b)

**Figure 20.** The rotation error and translation error: (**a**) rotation error $E_{rot}$; (**b**) translation error $E_{trans}$.

However, because the term $\|t_{true}\|$ in Equation (24) is a variable, when the distance between the camera and guidance lamps along the *z*-axis became smaller, it would also become smaller. It was difficult to reflect the change in the translation error with a change in the distance between the camera and guidance lamps along the *z*-axis. Therefore, the equation is modified as follows:

$$E_{transc} = \|t - t_{true}\| / \|t_r\| \times 100\%, \tag{25}$$

where $t_r$ is the range of $t$. The results are shown in Figure 21, which shows that the translation error was not more than 8%, and the smaller the distance between the camera and guidance lamps along the *z*-axis, the smaller the translation error.

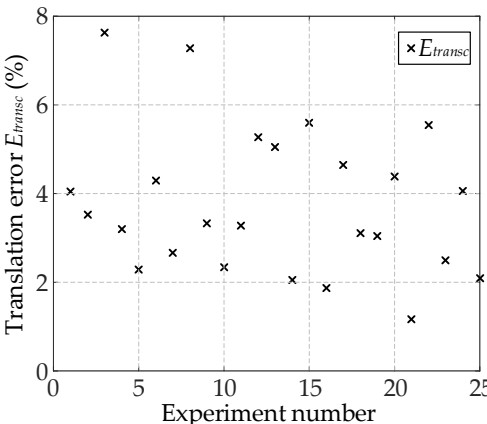

**Figure 21.** Modified translation error $E_{transc}$ from various experiments.

### 5.4. Comparison of Proposed Method and OI Method

*Slovepnp*, a classical function in *Opencv* for poses and positions estimation, is used to solve PnP problems. The OI method in the *Slovepnp* function has the highest accuracy. Therefore, we compare the PPARD estimation and OI methods. The guidance feature extraction was carried out using the method proposed in this study, and the PPARD estimation results are presented in Table 2.

**Table 2.** Statistical indicators of PPARD estimation results.

| Evaluating Indicators | DOF | Proposed Method | OI |
|---|---|---|---|
| | $\theta_x$ (°) | 0.42 | 3.77 |
| | $\theta_y$ (°) | 0.45 | 3.95 |
| Mean error of RRARD | $\theta_z$ (°) | 0.34 | 1.22 |
| | $l_x$ (mm) | 45.76 | 163.60 |
| | $l_y$ (mm) | 36.07 | 191.88 |
| | $l_z$ (mm) | 61.80 | 89.97 |
| | $\theta_x$ (°) | 0.91 | 6.59 |
| | $\theta_y$ (°) | 0.90 | 5.92 |
| Maximum error of PPARD | $\theta_z$ (°) | 0.89 | 3.03 |
| | $l_x$ (mm) | 76.67 | 294.50 |
| | $l_y$ (mm) | 69.20 | 274.30 |
| | $l_z$ (mm) | 132.25 | 131.65 |
| | $\theta_x$ (°) | 0.49 | 4.28 |
| | $\theta_y$ (°) | 0.53 | 4.36 |
| Root mean squared error of PPARD | $\theta_z$ (°) | 0.41 | 1.47 |
| | $l_x$ (mm) | 49.89 | 206.35 |
| | $l_y$ (mm) | 40.73 | 219.20 |
| | $l_z$ (mm) | 71.30 | 98.85 |

### 5.5. Analysis of Guidance Feature Extraction

5.5.1. Success Rate of Guidance Feature Extraction

The $D_{recovery}$ dataset was used to study the guidance feature extraction. It contained 2306 images of active underwater guidance lamps. $D_{recovery}$ was collected by the SIA-3 AUV (shown in Figure 22) in shallow-water field experiments using the lamps and docking station deployed in [14]. The SIA-3 AUV had a diameter of 384 mm, a length of 5486 mm, and a weight of 1500 kg in air. The maximum distance between the AUV and the lamps was approximately 8 m in the experiments. The extraction was regarded as a successful extraction if the number of detected lamps was equal to the number of predefined lamps; otherwise, it was regarded as a failed extraction.

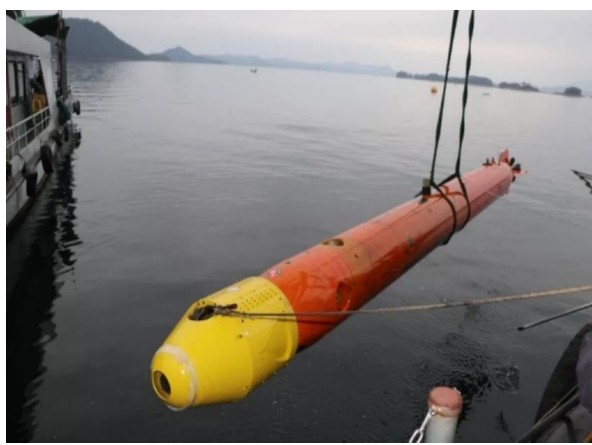

**Figure 22.** The SIA-3 AUV.

The guidance images (dataset $D_{experiment}$) collected in our experiments were also used to test the image processing algorithm. The results are presented in Table 3. The maximum distance between the camera and the lamps was 4 m in the experiments. The success rate of guidance feature extraction without image preprocessing was 68.89% for $D_{recovery}$, and was improved to 87.99% when image preprocessing was carried out. Thus, there was an improvement of approximately 19%. At the same time, the rate of fake guidance lamps extracted from the guidance image decreased to 0.39%. Therefore, image preprocessing was indispensable for feature extraction.

**Table 3.** Results of guidance features for datasets $D_{recovery}$ and $D_{experiment}$.

| Database | Without Image Preprocessing | | With Image Preprocessing | |
| --- | --- | --- | --- | --- |
| | Successful Extraction (%) | Fake Guidance Lamps Existing in the Failed Extraction (%) | Successful Extraction (%) | Fake Guidance Lamps Existing in Failed Extraction (%) |
| $D_{recovery}$ | 68.89 | 8.58 | 87.99 | 0.39 |
| $D_{experiment}$ | 100 | 0 | 100 | 0 |

5.5.2. Comparison of Adaptive Threshold Image Binarization and Classical Binarization Methods

Next, four GFEBIB methods were compared with the method proposed in this study. The four methods were fixed thresholding, Otsu thresholding, histogram-based adaptive thresholding (HATS), and histogram-based statistical thresholding (HSTS).

Fixed thresholding uses a guidance feature extraction threshold. The value of points higher than the threshold in the guidance image are assigned a value of 1, and all the other points are assigned a value 0. The guidance features are extracted using morphological processing. Through testing data set $D_{recovery}$, this method could obtain the highest success rate of guidance feature extraction, when the threshold was 233.5.

The basic idea of Otsu thresholding is to use a threshold to divide the data in the image into two categories. In one category, the intensity of the image pixels is less than this threshold, and in the other category, the intensity of the image pixels is greater than or equal to this threshold. If the variance of the intensity of the pixels in the two classes is greater, the obtained threshold is the best threshold. Then, the guidance image can be binarized to extract guidance features using the threshold.

The HATS is a guidance feature extraction method proposed in reference [25] aiming at the docking problem between the AUV and the docking station. It takes the value of the second largest peak for the guidance image histogram as the binarization threshold.

The HSTS is used to count the proportion of pixels in the histogram that are less than the threshold $t_h$. When the pixel proportion reaches $a$, the corresponding threshold $t_h$ is selected as the binarization threshold of the guidance image. In this study, parameter $a$ was selected as 0.93. At this time, the guidance feature extraction of data set $D_{recovery}$ achieved the highest success rate.

Figure 23 shows the guidance feature extraction results of the five methods for an example guidance image. The comparison results are presented in Table 4. It can be seen from the results that the guidance feature extraction success rate of the proposed method was the highest among the five methods for $D_{recovery}$ and $D_{experiment}$. Therefore, it was more conducive to the PPARD estimation and visual guidance of AUV docking.

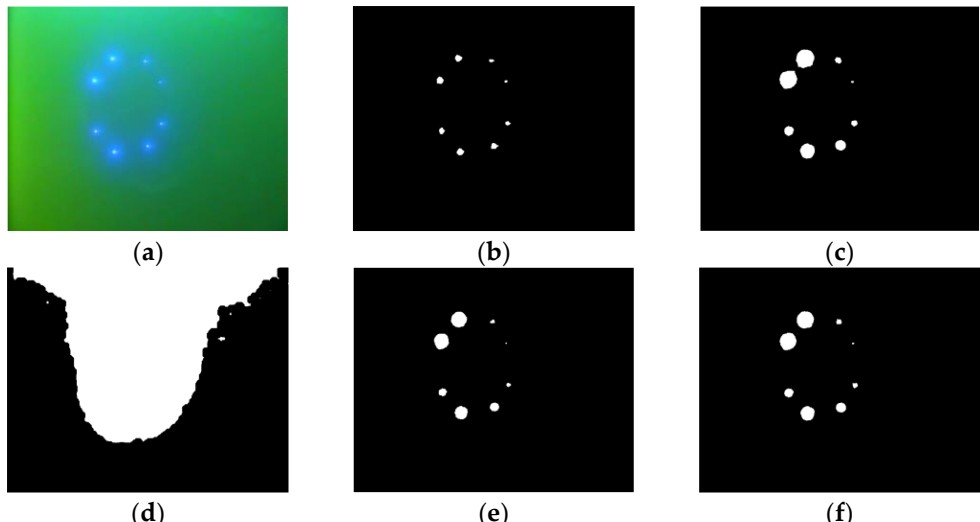

**Figure 23.** The guidance feature extraction results of five methods for an example guidance image: (**a**) original image; (**b**) extracted guidance features from the method proposed in this study; (**c**) extracted guidance features from fixed thresholding; (**d**) extracted guidance features from Otsu thresholding; (**e**) extracted guidance features from HATS; and (**f**) extracted guidance features from HSTS.

**Table 4.** Comparison of results for guidance feature extraction using five different methods.

| Data Set | Proposed Method | Fixed Thresholding | Otsu Thresholding | HATS | HSTS |
| --- | --- | --- | --- | --- | --- |
| $D_{recovery}$ | 87.99% | 35.94% | 2.29% | 34.47% | 34.78% |
| $D_{experiment}$ | 100% | 41.94% | 4.12% | 31.46% | 32.21% |

## 6. Conclusions and Discussion

Aiming at the docking problem between an AUV and a docking station, a new PPARD estimation method based on visual guidance feature extraction was proposed in this paper. The conclusions of this study are as follows:

(1) This study derived the imaging error function of the guidance features according to the principle of camera imaging. The model for estimating PPARD was built by optimizing the function to obtain a minimum value.

(2) The characteristics of the guidance images were analyzed. The guidance lamps were found to be strong blue point light sources. Based on this characteristic, an adaptive threshold binarization method for the guidance image was proposed in this study. The guidance features were extracted by combining image binarization and morphological processing.

(3) To improve the robustness of guidance feature extraction, this study proposed an enhancement method for guidance images. It became more obvious that the guidance lamps were strong point light sources with a certain pixel area in the enhanced image. In addition, the texture features of the guidance image were analyzed and used to segment the guidance target area. Image enhancement and segmentation could improve the failure of guidance feature extraction caused by changes in the photographing environment and pose/position of the camera.

(4) A PPARD estimation test platform was designed in this study. The experimental results of the PPARD estimation showed that the estimation accuracy of the proposed

method was better than that of the OI method. The absolute errors of the three rotation angle estimations were less than $1°$; the position estimation errors along the $x$- and $y$- axes were less than 80 mm; and the position estimation error along the $z$-axis was less than 140 mm. In the comprehensive evaluation index, the rotation error was less than $1.3°$ and the translation error was less than 8%. Furthermore, the feature extraction method proposed in this study was compared to fixed thresholding, Otsu thresholding, HATS, and HSTS using datasets $D_{recovery}$ and $D_{experiment}$. The results showed that the guidance feature extraction method proposed in this study yielded a better environmental adaptability and success rate.

The limitations and application extension need to be discussed.

(1) Limitations. This method requires that the brightness of the guidance lamps is similar. Its great difference can easily cause the failure of guidance feature extraction. Thus, the PPARD estimation cannot be realized. If one or two guidance lamps are too dark, the pixel value of the corresponding imaging position in the obtained guidance image is low. The adaptive threshold calculated using Equation (9) may be higher than the maximum pixel value of the too dark guidance lamps, resulting in the loss of guidance lamps in the binary image. If one or two guidance lamps are too bright, the area between adjacent guidance lamps may be overexposed, and the guidance lamps will be connected into one piece. It is impossible to extract all guidance lamps. All of the above will lead to the invalidation of the PPARD estimation method.

(2) Application extension. This method is proposed to aim toward the problem of AUV underwater docking guidance. It will be applied to the AUV docking operation in the future. Its main function is to estimate the PPARD. This method can be applied to other scenes that need to estimate the positions and poses of the AUV relative to the target objects through further research and development. For example, in the AUV cluster, AUVs can use this method to estimate the relative positions and poses between AUV groups by carrying guidance lamps. In addition, for oil field pipeline intervention operations, guidance lamps can be configured on the pipeline valve to estimate the positions and poses of the AUV relative to the valve, and then control the manipulator installed on the AUV to screw the valve.

In the future, the following studies are required.

(1) The prerequisite for the feature extraction of this method is that the four guidance lamps in the image cannot become a large over-exposed dot, and there must be a non-over-exposed region between the four guiding lamps. Camera parameters, the size of the docking station, and the power of guidance lamps all affect the success of feature extraction. The limited range of distance from the camera to the docking station and whether the algorithm is working well need to be analyzed for the specific docking application. In the future, this method will be used in the actual AUV docking scenarios. During docking, multiple experimental analyses will be required to determine the suitable distance for feature extraction and PPARD estimation.

(2) In the experiment results, the failure in the extraction of guidance lamps was mostly owing to the fact that the guidance lamp extraction had not received all the guidance lamps. Aiming at this type of extraction results, the PPARD estimation algorithm needs to be studied to ensure that it can also be carried out when the guide lamps are not extracted completely. In addition, there were pseudo guidance lamps in some failure extraction results. The image classification-based identification of guidance lamps and the proposal of pseudo guidance lamps need to be studied in the future.

(3) The vision-based PPARD estimation proposed in this study had a high accuracy. The following work for the docking problem of the AUV and docking station involves applying it to an AUV docking control. It can provide feedback information for docking control and help the AUV to realize docking with a high precision and high success rate. Furthermore, lake and ocean experiments for AUV docking will be carried out to test the performance of PPARD estimation and AUV docking control.

**Author Contributions:** Conceptualization, F.L.; methodology, F.L.; validation, F.L.; formal analysis, F.L.; investigation, F.L., H.X., and K.S.; writing—original draft preparation, F.L.; writing—review and editing, H.X. and X.W.; supervision, H.X.; funding acquisition, H.X. and K.S. All authors have read and agreed to the published version of the manuscript.

**Funding:** This research was funded by the Chinese Academy of Sciences Strategic Pilot Science and Technology Special, grant number XDA22040103, The R & D Projects in Key Areas of Guangdong Province, grant number 2020B1111010004, and Laboratory Development Foundation of Shenyang Institute of Automation, Chinese Academy of Sciences, grant number SIA2021ZZBS03.

**Data Availability Statement:** Not applicable.

**Conflicts of Interest:** The authors declare no conflict of interest. The funders had no role in the design of the study; in the collection, analyses, or interpretation of data; in the writing of the manuscript; or in the decision to publish the results.

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
