# Peer review of "Estimation of Positions and Poses of Autonomous Underwater Vehicle Relative to Docking Station Based on Adaptive Extraction of Visual Guidance Features"

_machines, doi:10.3390/machines10070571_

Round 1

Reviewer 1 Report

Please find attached the file for my comments. Thank you. 

Author Response

The response for the comments can be seen in the attached file.

Reviewer 2 Report

This paper is to estimate the pose and position based on adaptive extraction. Then, the algorithm is validated and verified via experiments to prove to be better than the classical methods. This algorithm is good application research. Here are my minor comments.

Section 2:
In eq. 5 and 7, the authors should present the suitable range of predicted pixel coordinates. What are the constraints of nonlinear programming of optimization techniques?

Section 3:
The blue component B is used to extract the guidance features. How clear is the original image that the author can apply the algorithm?

Section 5:
In 5.5.1, the success rate of guidance feature extraction is presented in Table 4. How far is the distance from the camera to the object that the algorithm is working well? The success rate should be recommended in a limited range.
